# Toward Equation of Motion for Deep Neural Networks: Continuous-time Gradient Descent and Discretization Error Analysis

**Taiki Miyagawa**
NEC Corporation, Japan
`miyagawataik@nec.com`

## Abstract

We derive and solve an "Equation of Motion" (EoM) for deep neural networks (DNNs), a differential equation that precisely describes the discrete learning dynamics of DNNs. Differential equations are continuous but have played a prominent role even in the study of discrete optimization (gradient descent (GD) algorithms). However, there still exist gaps between differential equations and the actual learning dynamics of DNNs due to *discretization error*. In this paper, we start from gradient flow (GF) and derive a counter term that cancels the discretization error between GF and GD. As a result, we obtain *EoM*, a continuous differential equation that precisely describes the discrete learning dynamics of GD. We also derive discretization error to show to what extent EoM is precise. In addition, we apply EoM to two specific cases: scale- and translation-invariant layers. EoM highlights differences between continuous-time and discrete-time GD, indicating the importance of the counter term for a better description of the discrete learning dynamics of GD. Our experimental results support our theoretical findings.

## 1 Introduction

Let us first explain our primary motivation for the present paper. In *physics*, one of the fundamental goals is to predict the dynamics of matter and its fundamental constituents. Specifically, "predict" here means to construct differential equations that best describe the physical system under consideration and to solve them. Such differential equations are called *Equations of Motion* (EoM). An interesting question here may be "What is the EoM for deep neural networks (DNNs)?" That is, to what extent can we predict the discrete learning dynamics of DNNs by constructing differential equations? This is our research question.

Differential equations have played a prominent role in studying discrete optimization (gradient descent (GD) algorithms), although they are continuous [1, 2, 3, 4, 5, 6, 7, 8, 9, 10, 11, 12, 13, 14, 15, 16, 17, 18, 19, 20]. In the context of deep

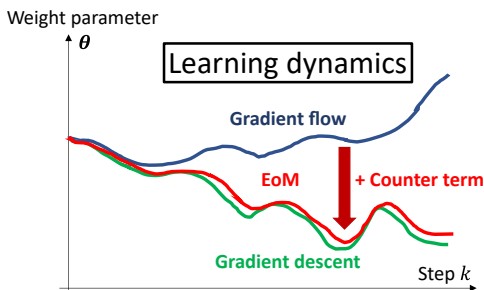

Figure 1: **Our approach.** GF fails in describing the learning dynamics of GD due to *discretization error*. Our counter term approach successfully cancels the discretization error between GF and GD and hence allows for a reliable analysis of GD.

learning, gradient flow (GF) and stochastic differential equations (SDEs) are used to analyze (stochastic) gradient descent ((S)GD). Research targets include: convergence [6, 7, 8, 12, 13, 9, 14, 17],

36th Conference on Neural Information Processing Systems (NeurIPS 2022).

stability of optimization [19], optimization with constraints [19], convergent states [17, 20], flatness of loss landscapes [17], empirical risk bounds [15], and online PCA [11]. Various techniques for continuous analysis have been imported to the analysis of discrete GD algorithms.

However, there still exist gaps between differential equations and actual learning dynamics due to *discretization error*, which is the main interest of the present paper and is often missing in the literature above. To be specific, we focus on GF $\dot{\boldsymbol{\theta}}(t) = -\boldsymbol{g}(\boldsymbol{\theta}(t))$ as a continuous approximation of GD $\boldsymbol{\theta}_{k+1} = \boldsymbol{\theta}_k - \eta\boldsymbol{g}(\boldsymbol{\theta}(t))$, where $\boldsymbol{\theta}(t) \in \mathbb{R}^d$ and $\boldsymbol{\theta}_k \in \mathbb{R}^d$ are the weight parameters of a DNN at time $t \in \mathbb{R}$ and step $k \in \mathbb{Z}$, respectively, and $\boldsymbol{g} \in \mathbb{R}^d$ is a gradient vector. $\eta \in \mathbb{R}$ is a learning rate and is regarded as the discretization step size when GF is discretized with the Euler method [21]: $\dot{\boldsymbol{\theta}}(t = k\eta) \coloneqq \frac{\boldsymbol{\theta}_{k+1}-\boldsymbol{\theta}_k}{\eta}$. Due to this approximation, discretization error (or "continuation error") is introduced, and thus GF cannot fully explain the dynamics of GD. For instance, we show that according to GF, the weight norm of a scale-invariant layer collapses to zero when we use weight decay, while GD does not show such behavior (Section 5.1).

To fill the critical gap between GF and GD, we propose modifying GF to describe the learning dynamics of GD more precisely; i.e., we add a counter term $\boldsymbol{\xi} \in \mathbb{R}^d$ to the gradient $\boldsymbol{g}$ of GF that cancels the discretization error (Figure 1). This idea is motivated by backward error analysis in numerical analysis [21]. We derive a functional integral equation that determines the counter term and solve it (Section 3). As a result, we obtain a more reliable differential equation, called *EoM* here, that describes the discrete learning dynamics of GD. Using the counter term, we derive the leading order of discretization error (Section 4.1) to show to what extent GF and EoM are precise in describing GD's dynamics. This point is often missed in the literature on the continuous approximation of discrete GD algorithms [22, 23, 24, 11, 25, 26, 27, 28]. We further derive a sufficient condition for learning rates for the discretization error to be small (Section 4.2). We show that EoM well explains empirical results.

Furthermore, to show the benefits of EoM, we apply it to two specific cases: scale-invariant layers [29, 30] and translation-invariant layers [31, 32] (Section 5). For scale-invariant layers, we show that a better description of GD's discrete dynamics requires modifications to the decay rate of weight norms that is previously derived in the continuous regime (SDEs) [33]. In addition, we show that EoM successfully reproduces the limiting dynamics ($t \to \infty$) of weight norms and angular update [34] that are previously derived in the discrete regime, while GF cannot reproduce this result. For translation-invariant layers, we show that EoM rather than GF dramatically matches empirical results, indicating the importance of the counter term. To the best of our knowledge, no study analyzes the temporal evolution of translation-invariant layers except for [31] and [32], where only the sum of weights is their focus, while we derive the dynamics of the whole weights.

Our contribution is four-fold. Our code[1] and detailed experimental results are given as supplementary materials.

1. To fill the critical gap between GF and GD, we derive a counter term for GF that cancels the discretization error, and as a result, we obtain EoM, a continuous differential equation that precisely describes the discrete learning dynamics of GD.

2. To show to what extent GF and EoM are precise in describing discrete GD dynamics, we derive the leading order of discretization error, as is often missed in the literature on the continuous approximation of discrete GD algorithms. We further derive a sufficient condition for learning rates for the discretization error to be small.

3. We apply EoM to two specific cases: scale-invariant layers and translation-invariant layers, indicating the importance of the counter term for a better description of the discrete learning dynamics of GD.

4. Our experimental results support our theoretical findings.

Our work is the first step toward answering this research question: to what extent can we predict the discrete learning dynamics of DNNs by constructing differential equations (EoM for DNNs)? Also, our work helps researchers import continuous analysis to the discrete analysis of GD algorithms. In this sense, our work bridges discrete and continuous analyses of GD algorithms.

---

[1]See Supplementary Materials at `https://openreview.net/forum?id=qq84D17BPu` .

# 2 Related Work

The idea of approximating discrete-time stochastic algorithms with continuous equations dates back to stochastic approximation theory [1, 2, 3, 4, 5]. Their primary focus is convergence analysis for discrete-time algorithms, while our focus is to predict the learning dynamics (temporal evolution) of weight parameters, such as the decay rates of weight norms and effective learning rate of scale-invariant layers. Our idea of the counter term is inspired by the backward error analysis developed for numerical analysis [35]. This idea is now used to analyze discrete optimization [22, 23, 24, 11, 25, 26, 27, 28]. [18] is a pioneering work on discretization error analysis between GF and GD that is based on the numerical analysis of the Euler method [21]. They derive a sufficient condition for learning rates for the discretization error to be small. This analysis is based on a bound (inequality), while we derive an explicit relationship between learning rates and discretization error as an equality.

Neural mechanics and Noether's learning dynamics [31, 32] provide a solution to a part of the aforementioned problem: to what extent can we predict the learning dynamics of DNNs by constructing differential equations? They derive (the breaking of) conservation laws of weight parameters using differential equations and provide the temporal evolution of the conserved quantities. The present work is inspired by these studies but has crucial differences: 1) our focus is on the temporal evolution of all of the network parameters, not only the conserved quantities, 2) the gradient's correction for canceling the discretization error is not limited to the first order, but all orders, and 3) the discretization error is explicitly provided in the present paper. See Appendix G for more related studies.

# 3 Equation of Motion for Deep Neural Networks

In the following sections, we define *EoM* by modifying GF (Section 3.1). We show that the counter term satisfies a functional integral equation (Section 3.2), and then we solve it (Section 3.3).

## 3.1 Our Approach and Definitions

We begin with a simple idea: add a counter term to GF to cancel discretization error, i.e.,

$$\dot{\boldsymbol{\theta}}(t) = -\boldsymbol{g}(\boldsymbol{\theta}(t)) - \eta\boldsymbol{\xi}(\boldsymbol{\theta}(t)), \tag{1}$$

where $\boldsymbol{\theta}(t) \in \mathbb{R}^d$ is the vectorized weight parameters of a DNN at time $t \in \mathbb{R}$, $d \in \mathbb{N}$ is the dimension of the weight, and $\dot{\boldsymbol{\theta}}(t)$ denotes $d\boldsymbol{\theta}(t)/dt$. Gradient $\boldsymbol{g}(\boldsymbol{\theta}(t))$ is defined as $\boldsymbol{g}(\boldsymbol{\theta}(t)) := \nabla f(\boldsymbol{\theta}(t)) + \lambda\boldsymbol{\theta}(t)$, which consists of a loss function $f(\boldsymbol{\theta}(t))$ and weight decay term $\lambda\boldsymbol{\theta}(t)$, where $\lambda > 0$ controls the strength of weight decay. $\eta > 0$ is a small learning rate, and $\boldsymbol{\xi}(\boldsymbol{\theta}(t)) \in \mathbb{R}^d$ is the counter term. Throughout this paper, we assume all functions are sufficiently smooth. We call Equation (1) the *Equation of Motion (EoM)* for DNNs, or simply EoM.

Our aim is to find $\boldsymbol{\xi}$ that makes Equation (1) more reliable to precisely approximate GD $\boldsymbol{\theta}_{k+1} = \boldsymbol{\theta}_k - \eta\boldsymbol{g}(\boldsymbol{\theta}_k)$, where $\boldsymbol{\theta}_k \in \mathbb{R}^d$ is the weight at step $k \in \mathbb{Z}_{\geq 0}$. To do so, we first define the *discretization error* between GF (1) and GD at step $k$:

$$\boldsymbol{e}_k := \boldsymbol{\theta}(k\eta) - \boldsymbol{\theta}_k \ \in \mathbb{R}^d \tag{2}$$

and find $\boldsymbol{\xi}$ that makes $\boldsymbol{e}_k$ small. Throughout this paper, we use the standard Euler method to discretize GF: $\dot{\boldsymbol{\theta}}(t) \fallingdotseq (\boldsymbol{\theta}(t+\eta) - \boldsymbol{\theta}(t))/\eta$ and $t = k\eta$; thus, $\eta$ is identified with the discretization step size.

## 3.2 How to Determine Counter Term

We show that the leading order of $\boldsymbol{e}_k$ with respect to $\eta$ is controlled by the counter term (Theorem 3.2), and as a result, the counter term is determined via a functional integral equation (Equation (6)).

Our first theorem shows what the counter term should cancel.

**Theorem 3.1** (Recursive formula for discretization error)**.** *Discretization error $\boldsymbol{e}_k$ satisfies:*

$$\boldsymbol{e}_{k+1} - \boldsymbol{e}_k = -\eta\big(\boldsymbol{g}(\boldsymbol{\theta}(k\eta)) - \boldsymbol{g}(\boldsymbol{\theta}(k\eta) - \boldsymbol{e}_k)\big) + \eta^2 \int_0^1 ds\ddot{\boldsymbol{\theta}}(\eta(k+s))(1-s) - \eta^2\boldsymbol{\xi}(\boldsymbol{\theta}(k\eta)) \tag{3}$$

$$=: -\eta\left(\boldsymbol{g}(\boldsymbol{\theta}(k\eta)) - \boldsymbol{g}(\boldsymbol{\theta}(k\eta) - \boldsymbol{e}_k)\right) + \boldsymbol{\Lambda}(\boldsymbol{\theta}(k\eta)). \tag{4}$$

Here, we defined $\mathbf{\Lambda}(\boldsymbol{\theta}(k\eta)) := \eta^2 \int_0^1 ds \ddot{\boldsymbol{\theta}}(\eta(k+s))(1-s) - \eta^2 \boldsymbol{\xi}(\boldsymbol{\theta}(k\eta)) \in \mathbb{R}^d$. The proof is based on Taylor's theorem and is given in Appendix A.1. The right-hand side of Equation (3) tells us that the counter term (third term) should cancel the first and second terms. However, the following theorem states that the first term gives only subleading contributions with respect to $\eta$.

**Theorem 3.2** (Leading order of discretization error). *Suppose that $\mathbf{\Lambda}(\boldsymbol{\theta}(k\eta)) = O(\eta^\gamma)$ and $\boldsymbol{e}_0 = O(\eta^\gamma)$ for some $\gamma > 0$. Then $\boldsymbol{e}_k = O(\eta^\gamma)$ and $-\eta(\boldsymbol{g}(\boldsymbol{\theta}(k\eta)) - \boldsymbol{g}(\boldsymbol{\theta}(k\eta) - \boldsymbol{e}_k)) = O(\eta^{\gamma+1})$. Therefore, the first term in the right-hand side of Equation (3) is negligible compared with $\mathbf{\Lambda}$:*

$$\boldsymbol{e}_{k+1} = \boldsymbol{e}_k + \mathbf{\Lambda}(\boldsymbol{\theta}(k\eta)) - \eta(\boldsymbol{g}(\boldsymbol{\theta}(k\eta)) - \boldsymbol{g}(\boldsymbol{\theta}(k\eta) - \boldsymbol{e}_k))$$
$$= \boldsymbol{e}_k + \mathbf{\Lambda}(\boldsymbol{\theta}(k\eta)) + O(\eta^{\gamma+1}) \qquad (k = 0, 1, 2, ...). \tag{5}$$

The proof is by induction and given in Appendix A.2. Therefore, the leading order of discretization error is $O(\eta^\gamma)$ and given by:

$$\mathbf{\Lambda}(\boldsymbol{\theta}(k\eta)) = O(\eta^\gamma) \iff \int_0^1 ds \, \ddot{\boldsymbol{\theta}}(\eta(k+s))(1-s) - \boldsymbol{\xi}(\boldsymbol{\theta}(k\eta)) = O(\eta^{\gamma-2}). \tag{6}$$

This is a functional equation of $\boldsymbol{\xi}$ because $\ddot{\boldsymbol{\theta}}(t)$ contains $\boldsymbol{\xi}$ via Equation (1). A solution to Equation (6) for a large $\gamma$ gives a small $\mathbf{\Lambda}$ and thus gives a small $\boldsymbol{e}_k$ via Equation (5).

### 3.3 Solution to Equation 6

How can we solve Equation (6)? It is not easy to find an exact solution because Equation (6) is a functional integral equation [36, 37, 38, 39, 40]; therefore, we assume a power series solution with respect to $\eta$:

$$\boldsymbol{\xi}(\boldsymbol{\theta}(k\eta)) = \sum_{\alpha=0}^{\infty} \eta^\alpha \boldsymbol{\xi}_\alpha = \boldsymbol{\xi}_0(\boldsymbol{\theta}(k\eta)) + \eta \boldsymbol{\xi}_1(\boldsymbol{\theta}(k\eta)) + \eta^2 \boldsymbol{\xi}_2(\boldsymbol{\theta}(k\eta)) + \cdots . \tag{7}$$

In the following theorem, we successfully find a solution for *all* orders of $\eta$.

**Theorem 3.3** (Solution of Equation 6). *The solution to Equation (6) of form (7) is given by*

$$\boldsymbol{\xi}_\alpha(\boldsymbol{\theta}) = \tilde{\boldsymbol{\xi}}_\alpha(\boldsymbol{\theta}) := \sum_{i=2}^{\alpha+2} \sum_{k_1 + \cdots + k_i = \alpha - i + 2} \frac{(-1)^i}{i!} D_{k_1} \cdots D_{k_{i-1}} \Xi_{k_i} \tag{8}$$

*for $\alpha = 0, 1, 2, ...$, where we use differential operators (Lie derivatives) $\mathcal{D}_\alpha := \boldsymbol{\xi}_{\alpha-1}(\boldsymbol{\theta}) \cdot \nabla$ ($\alpha = 1, 2, ...$) and $\mathcal{D}_0 := \boldsymbol{g}(\boldsymbol{\theta}) \cdot \nabla$ and also defined $\Xi_\alpha(\boldsymbol{\theta}) := \boldsymbol{\xi}_{\alpha-1}(\boldsymbol{\theta})$ ($\alpha = 1, 2, ...$) and $\Xi_0(\boldsymbol{\theta}) := \boldsymbol{g}(\boldsymbol{\theta})$.*

The proof follows from the definition of the Lie derivative and is given in Appendix A.3. The first two orders of the solution are given by:

$$\tilde{\boldsymbol{\xi}}_0(\boldsymbol{\theta}) = \frac{1}{2}(\boldsymbol{g}(\boldsymbol{\theta}) \cdot \nabla)\boldsymbol{g}(\boldsymbol{\theta}) = \frac{1}{4}\nabla\|\boldsymbol{g}(\boldsymbol{\theta})\|^2 \tag{9}$$

$$\tilde{\boldsymbol{\xi}}_1(\boldsymbol{\theta}) = \frac{1}{2}(\tilde{\boldsymbol{\xi}}_0(\boldsymbol{\theta}) \cdot \nabla)\boldsymbol{g}(\boldsymbol{\theta}) + \frac{1}{6}(\boldsymbol{g}(\boldsymbol{\theta}) \cdot \nabla)\tilde{\boldsymbol{\xi}}_0 . \tag{10}$$

**Discussions.** As can be inferred from Equations (8–10), $\tilde{\boldsymbol{\xi}}_\alpha$ contains the $\alpha + 2_{\mathrm{nd}}$-order derivative of the loss function. Therefore, the higher-order counter terms cancel the higher-order smoothness of the discretization error.

Here, we note that Equation (8) can be found, e.g., in [35], as a higher-order backward error analysis. However, our derivation above has independent contributions: 1) we clarify that the counter term cancels the leading order of discretization error (Theorem 3.2), and 2) we find that the discretization error itself is also given by the counter term (Corollary 4.1 in the next section).

Equation (9) often appears in the literature on backward error analysis [21, 35] and its related topics in machine learning, e.g., [41, 23, 24, 27, 28, 31]. Typically, $\tilde{\boldsymbol{\xi}}_0$ is added to gradients of continuous equations (e.g., SDE) to close the gap between continuous equations and discrete algorithms (e.g., SGD) by canceling (at least first-order) discretization error. However, higher-order discretization error is neglected in these studies. In contrast, our solution (8) cancels *all* orders of discretization error.

# 4 Discretization Error

The question here is to what extent the continuous approximation (1, 8) is precise; this point is often missed in the literature on continuous approximation [22, 23, 24, 11, 25, 26, 27, 28]. In this section, we use the counter term (8) and quantify discretization error as a function of the loss function and its derivatives (Section 4.1). We find that our result well explains empirical results. We further derive a sufficient condition for learning rates for the discretization error to be small (Section 4.2).

## 4.1 Counter Term Gives Leading Order of Discretization Error

We show that the counter term gives the leading order of discretization error between GD vs. GF and EoM. The proof follows from Theorem 3.2 and 3.3 and is given in Appendix A.4.

**Corollary 4.1** (Leading order of discretization error is given by $\tilde{\boldsymbol{\xi}}_\alpha$). *Suppose that we use $\boldsymbol{\xi}$ up to $O(\eta^{\gamma-1})$, i.e., $\boldsymbol{\xi} = \tilde{\boldsymbol{\xi}}_0 + \eta\tilde{\boldsymbol{\xi}}_1 + \cdots + \eta^{\gamma-1}\tilde{\boldsymbol{\xi}}_{\gamma-1}$ for $\gamma \in \mathbb{Z}_{>0}$ ($\boldsymbol{\xi} := \mathbf{0}$ for $\gamma = 0$). Then,*

$$\boldsymbol{e}_{k+1} = \boldsymbol{e}_k + \boldsymbol{\Lambda}(\boldsymbol{\theta}(k\eta)) + O(\eta^{\gamma+3}) = \boldsymbol{e}_k + \eta^{\gamma+2}\tilde{\boldsymbol{\xi}}_\gamma + O(\eta^{\gamma+3}) \,. \tag{11}$$

First, Corollary 4.1 implies that the higher the orders of the counter term we use (large $\gamma$), the more precise EoM (1) is (small $\boldsymbol{e}_k$). Thus, GF ($\boldsymbol{\xi} = \mathbf{0}$) gives larger discretization error than EoM ($\boldsymbol{\xi} \neq \mathbf{0}$). Second, Corollary 4.1 gives the *equality* of the leading order of discretization error at *arbitrary* steps. This is not a *bound* [18] nor an *asymptotic* analysis ($k \to \infty$). Third, let us give an intuition by considering $\boldsymbol{\xi} = \mathbf{0}$ (GF). Then, Corollary 4.1 gives:

$$\boldsymbol{e}_{k+1} = \boldsymbol{e}_0 + \sum_{s=0}^{k} \frac{\eta^2}{2}(H(\boldsymbol{\theta}(s\eta)) + \lambda I)(\nabla f(\boldsymbol{\theta}(s\eta)) + \lambda\boldsymbol{\theta}(s\eta)) + O(\eta^3) \,, \tag{12}$$

where $H(\boldsymbol{\theta}) \in \mathbb{R}^{d \times d}$ is the Hessian of the loss function $f$ with respect to $\boldsymbol{\theta}$ and $I \in \mathbb{R}^{d \times d}$ is the identity matrix. Equation (12) suggests that 1) large learning rates lead to a large discretization error and 2) steep loss functions (along the trajectory) lead to a large discretization error.

**Empirical result.** We find Equation (12) well explains our empirical result. We compare Equation (12) (up to $O(\eta^2)$) with the actual discretization error of GD and GF in Figure 2. First, the gap between our theoretical prediction of discretization error (orange curve) and the actual discretization error (red curve) is small because the range of *relative* error ($\|\boldsymbol{e}_k\|/\|\boldsymbol{\theta}_k\|$) in this plot is only 0–0.01 (see also Figure 11 in Appendix F). Second, most of the discretization error for Theory (orange curve) and Experiment (red curve) is produced within the first 100 steps. We can understand this phenomenon with the help of Equation (12). It suggests that discretization error can be enhanced when the loss function is non-smooth along the learning trajectory, which is likely to occur at the beginning of training due to random initialization. Therefore, a large part of discretization error is produced in the early stage of training. Third, we see that most of the gap between Theory (orange curve)

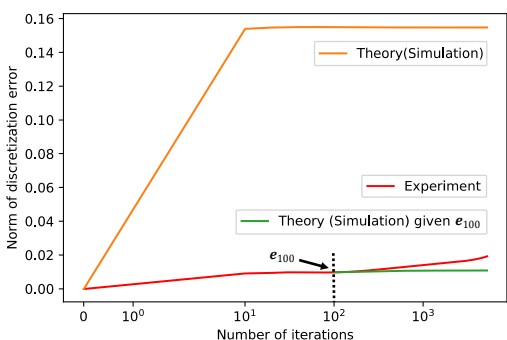

Figure 2: **Theoretical prediction of discretization error of GF and GD (Equation (12)) vs. actual discretization error of GF and GD.** The larning rate and weight decay are $10^{-2}$ and $10^{-2}$. See Appendix F.2 for more results and details. See Section 6 for experimental settings.

and Experiment (red curve) also comes from the first 100 steps; in fact, the green curve shows that there is a much smaller enhancement of the gap after the 100th step. The source of the gap is the higher-order term $O(\eta^3)$ in Equation (12). It consists of higher-order derivatives of the loss function (Theorem 8 and Corollary 4.1) and thus can be large when the loss function is non-smooth along the learning trajectory. Therefore, by the same logic as above, the early stage of training tends to produce a gap between Theory (orange curve) and Experiment (red curve).

## 4.2 Discretization Error Bounds

We provide a sufficient condition (an upper bound for $\eta$) for GF and EoM to follow GD up to a given step $k$, which helps us infer desired learning rates (step sizes) for the discretization error to be small. We first consider $\boldsymbol{\xi} = \mathbf{0}$ (GF).

**Corollary 4.2** (Learning rate bound for $\boldsymbol{\xi} = \mathbf{0}$). *Let $\boldsymbol{\xi} = \mathbf{0}$ and assume that $\boldsymbol{e}_0 = O(\eta^3)$. Let $\epsilon$ and $t$ be arbitrary positive numbers. If the step size satisfies*

$$\eta < \sqrt{\frac{\epsilon}{k}} \sqrt{\frac{2}{\max\limits_{0 \leq t' \leq t} \{||(H(\boldsymbol{\theta}(t')) + \lambda I)\boldsymbol{g}(\boldsymbol{\theta}(t'))||\}}} \,, \tag{13}$$

*for some $k \in \{1, 2, ..., \lfloor \frac{t}{\eta} \rfloor\}$, then the discretization error can be arbitrarily small:*

$$||\boldsymbol{e}_k|| < \epsilon + O(\epsilon^{\frac{3}{2}}) \,. \tag{14}$$

The proof follows from Equation (12) and is given in Appendix A.5. We see that 1) there is no guarantee that the discretization error is small unless the learning rate is sufficiently small, 2) we need small learning rates to keep the discretization error small for a long period, and 3) we need small learning rates to keep the discretization error small for non-smooth loss landscapes. This is consistent with our empirical results in Figure 3 and 4; in fact, 1) the discretization error blows up for a large learning rate ($\eta = 10^{-1}$ in Figure 3), 2) it increases as the number of steps increases (Figure 4), and 3) most of it is produced in the early phase of training, where the objective function tends to be non-smooth, and the gradients tend to be large.

We compare our bound (13) with a bound given in [18] because, to our knowledge, only [18] provides a bound for the step size with respect to discretization error in the context of deep learning. In [18], it is proved that in essence, $\eta \lesssim \epsilon/\beta_{t\epsilon}\gamma_{t\epsilon}c_t$, where $\beta_{t\epsilon}$ and $\gamma_{t\epsilon}$ measure the non-smoothness of the loss function, and $c_t$ depends on the spectrum of the Hessian. These factors are hard to compute analytically unless the loss function and network are simple, but the qualitative behavior of this bound is the same as ours (13); i.e., both bounds become tight when the loss function is non-smooth.

We also derive a learning rate bound for $\boldsymbol{\xi} = \tilde{\boldsymbol{\xi}}_0$ (EoM) and the full statement is given in Corollary A.1 in Appendix A.6, which states that if $\eta < O(\sqrt[3]{\frac{\epsilon}{k}})$, then $||\boldsymbol{e}_k|| < \epsilon + O(\epsilon^{\frac{4}{3}})$. Therefore, larger step sizes are now allowed compared with Corollary 4.2 (GF) because of the non-zero counter term. Furthermore, we can show larger bounds for higher-order counter terms in a similar way.

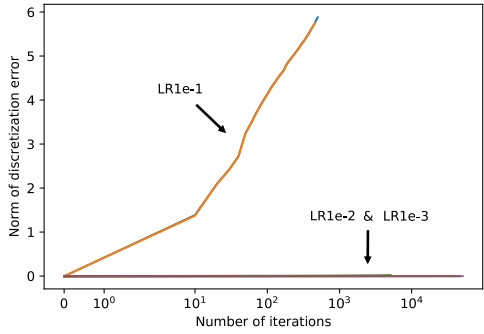
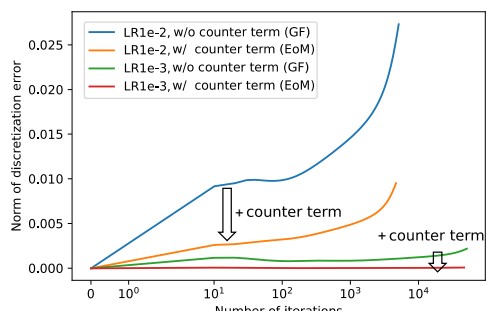

Figure 3: **Discretization error explodes for large learning rate** ($10^{-1}$). LR means learning rate. Weight decay is $10^{-3}$. Curves include both GF and EoM. Relative discretization error is also shown in Appendix F. See Section 6 for experimental settings.

Figure 4: **Discretization error of GF and EoM.** Figure 3 is magnified. The counter term reduces discretization error as expected, and smaller learning rates give smaller discretization errors.

## 5  Application: Scale- and Translation-invariant Layers

To show the benefits of EoM, we finally apply our theory to two specific cases: scale-invariant layers [29, 30] and translation-invariant layers [31, 32]. Additionally, Appendix B provides an application

to broken conservation laws [31]. In the following, we simply focus on $\boldsymbol{\xi} = \mathbf{0}$ and $\boldsymbol{\xi} = \tilde{\boldsymbol{\xi}}_0$ to analyze the differences between $\boldsymbol{\xi} = \mathbf{0}$ and $\boldsymbol{\xi} \neq \mathbf{0}$.

**Definitions** Let us first introduce our notation. A transformation $\psi$ of $\boldsymbol{\theta} \in \mathbb{R}^d$ with parameter $\alpha \in \mathbb{R}$ is said to be a *symmetry transformation* of loss function $f$ if $f(\psi(\boldsymbol{\theta}, \alpha)) = f(\boldsymbol{\theta})$. $\mathbb{1}_{\mathcal{A}} \in \{0, 1\}^d$ denotes the indicator vector of subspace $\mathcal{A} \subset \mathbb{R}^d$ (e.g., $\mathcal{A}$ is a linear layer in the DNN). For a scalar $\alpha \in \mathbb{R}$, we define $\alpha_{\mathcal{A}} := \alpha \mathbb{1}_{\mathcal{A}} + \mathbb{1}_{\mathcal{A}^c} \in \mathbb{R}$, where $\mathcal{A}^c$ is the complement of $\mathcal{A}$. For a vector $\boldsymbol{\theta} \in \mathbb{R}^d$, we define $\boldsymbol{\theta}_{\mathcal{A}} := \boldsymbol{\theta} \odot \mathbb{1}_{\mathcal{A}} \in \mathbb{R}^d$, where $\odot$ is the Hadamard element-wise product. For the gradient operator $\nabla = (\partial/\partial\theta_1, ..., \partial/\partial\theta_d)^\top$, we define $\nabla_{\mathcal{A}} := \mathbb{1}_{\mathcal{A}} \odot \nabla$. We also define $r_{\mathcal{A}} := ||\boldsymbol{\theta}_{\mathcal{A}}||$ and $\hat{\boldsymbol{\theta}}_{\mathcal{A}} := \boldsymbol{\theta}_{\mathcal{A}}/r_{\mathcal{A}}$.

## 5.1 Learning Dynamics of Scale-invariant Layers

In this section, we focus on scale-invariant layers. A scale-invariant layer $\mathcal{A}$ is defined as a subspace that is invariant under the scale transformation $\psi(\boldsymbol{\theta}, \alpha) := \alpha_{\mathcal{A}}\boldsymbol{\theta} = \alpha\boldsymbol{\theta}_{\mathcal{A}} + \boldsymbol{\theta}_{\mathcal{A}^c}$ ($\alpha > 0$). For example, a linear layer immediately before a batch normalization layer is scale-invariant. We see that for a better description of GD's discrete dynamics, we need modifications to the decay rate of $r_{\mathcal{A}}$ that is previously derived in the continuous regime [33]. In addition, we show that EoM successfully reproduces the limiting dynamics of $r_{\mathcal{A}}$ and *angular update* [34] at $t \to \infty$ that are previously derived in the discrete regime, while GF cannot. In Appendix C, we additionally show that there are crucial differences between GD and GF via the *effective learning rate* of scale-invariant layers [29, 42, 30, 43, 44, 33, 45, 34, 46, 47].

**EoM for $r$** We construct the EoM for $r_{\mathcal{A}}$ (the EoM for $\hat{\boldsymbol{\theta}}_{\mathcal{A}}$ is given in Appendix C for completeness).

**Theorem 5.1** (EoM for $r_{\mathcal{A}}$ and solution). *EoM (1) gives $\dot{r}_{\mathcal{A}}^2(t) = -2\lambda r_{\mathcal{A}}^2(t) - 2\eta\,\boldsymbol{\theta}_{\mathcal{A}}(t) \cdot \boldsymbol{\xi}(\boldsymbol{\theta}(t))$. Specifically, this is equivalent to:*

$$\dot{r}_{\mathcal{A}}^2(t) = -2\lambda r_{\mathcal{A}}^2(t) \iff r_{\mathcal{A}}^2(t) = r_{\mathcal{A}}^2(0)e^{-2\lambda t} \tag{15}$$

*for $\boldsymbol{\xi} = \mathbf{0}$ (GF) and*

$$\dot{r}_{\mathcal{A}}^2(t) = -2(\lambda + \frac{\eta\lambda^2}{2})r_{\mathcal{A}}^2(t) + \frac{\eta}{r_{\mathcal{A}}^2(t)}||\nabla_{\mathcal{A}}f(\hat{\boldsymbol{\theta}}_{\mathcal{A}}(t) + \boldsymbol{\theta}_{\mathcal{A}^c}(t))||^2 \tag{16}$$

$$\iff r_{\mathcal{A}}^2(t) = r_{\mathcal{A}}^2(0)e^{-2\lambda(1+\frac{\eta\lambda}{2})t} + \eta\int_0^t d\tau e^{-2\lambda(1+\frac{\eta\lambda}{2})(t-\tau)}\frac{||\nabla_{\mathcal{A}}f(\hat{\boldsymbol{\theta}}_{\mathcal{A}}(\tau) + \boldsymbol{\theta}_{\mathcal{A}^c}(\tau))||^2}{r_{\mathcal{A}}^2(\tau)} \tag{17}$$

*for $\boldsymbol{\xi} = \tilde{\boldsymbol{\xi}}_0$ (EoM).*

The proof is based on Equations (1, 9) and given in Appendix A.7. Equation (15) gives $r_{\mathcal{A}}^2(k\eta) = r_{\mathcal{A}}^2(0)e^{-2\eta\lambda k}$ ($k \in \mathbb{Z}_{\geq 0}$) at discretization; therefore, $\eta\lambda$ is regarded as the decay rate of $r_{\mathcal{A}}$ (*intrinsic learning rate* [33]). This is originally discussed in the continuous regime (SDE) [33]; however, we find that for a better description of the discrete dynamics of GD, the decay rate needs to be modified from $\eta\lambda$ to $\eta\lambda(1 + \frac{\eta\lambda}{2})$ (see the exponent of Equation (17)). This means that $r_{\mathcal{A}}$ in GD decays faster than expected from a naive continuous dynamics (GF (15) and SDE [33]). See Appendix G for higher-order corrections.

**Limiting dynamics.** We next derive the limiting dynamics ($t \to \infty$) of $r_{\mathcal{A}}$.

**Corollary 5.1** ($r_{\mathcal{A}}$ at equilibrium). *When $\boldsymbol{\xi} = \mathbf{0}$ (GF), $r_{\mathcal{A}}$ collapses to zero as $t \to \infty$. When $\boldsymbol{\xi} = \tilde{\boldsymbol{\xi}}_0$ (EoM), assume that there exist two constants $r_{\mathcal{A}*} \geq 0$ and $c_* \geq 0$ such that $r_{\mathcal{A}}(t) \xrightarrow{t\to\infty} r_{\mathcal{A}*}$ and $||\nabla_{\mathcal{A}}f(\hat{\boldsymbol{\theta}}_{\mathcal{A}}(t) + \boldsymbol{\theta}_{\mathcal{A}^c}(t))|| \xrightarrow{t\to\infty} c_*$. Then $r_{\mathcal{A}*}^2 = \sqrt{\frac{\eta}{2\lambda+\eta\lambda^2}}c_*$.*

The proof follows from Theorem 5.1 and is given in Appendix A.8. The non-zero counter term successfully reproduces $r_{\mathcal{A}*}^2 \sim \sqrt{\eta/2\lambda}\,c_*$ [29, 34], which is originally derived in the discrete regime (SGD), although our approach is continuous (EoM (1)). Without the counter term, we cannot explain this behavior because GF gives $r_{\mathcal{A}}(t) \xrightarrow{t\to\infty} 0 (\neq \sqrt{\eta/2\lambda}\,c_*)$.

We next derive the limiting dynamics of *angular update* [34], which is designed to measure the temporal evolution of scale-invariant networks. It is originally defined in the discrete regime:

$\cos \Delta_k := \hat{\boldsymbol{\theta}}_{\mathcal{A}k} \cdot \hat{\boldsymbol{\theta}}_{\mathcal{A}k+1}$, where $\hat{\boldsymbol{\theta}}_{\mathcal{A}k} := \frac{\mathbb{1}_{\mathcal{A}} \odot \boldsymbol{\theta}_k}{||\mathbb{1}_{\mathcal{A}} \odot \boldsymbol{\theta}_k||}$. That is, $\Delta_k$ represents a single-step angular change in the weight parameters of the scale-invariant layers $\mathcal{A}$. In the continuous regime, we can define $\cos \Delta(t) := \hat{\boldsymbol{\theta}}_{\mathcal{A}}(t) \cdot \hat{\boldsymbol{\theta}}_{\mathcal{A}}(t+\eta)$.

**Corollary 5.2** ($\Delta(t)$ at equilibrium). *Let us use $\boldsymbol{\xi} = \tilde{\boldsymbol{\xi}}_0$. Suppose that the assumptions in Corollary 5.1 are satisfied. The angular update at equilibrium, denoted by $\Delta_*$, is given by $\cos \Delta_* = \frac{1-\eta\lambda}{1-\eta^2\lambda^2/2} + O(\eta^3)$, and thus, $\Delta_* = \sqrt{2\eta\lambda} + O((\eta\lambda)^{3/2})$.*

The proof is based on Corollary 5.1 and is given in Appendix A.10. EoM successfully reproduces $\Delta_* \sim \sqrt{2\eta\lambda}$ [34], which is originally derived in the discrete regime (SGD), although EoM is continuous itself. On the other hand, GF cannot explain the limiting dynamics of $\Delta(t)$ because when $\boldsymbol{\xi} = \mathbf{0}$, $r(t)$ goes to zero as $t \to \infty$ (Equation (15)), and thus, $\cos \Delta(t) = \frac{\boldsymbol{\theta}_{\mathcal{A}}(t)}{r_{\mathcal{A}}(t)} \cdot \frac{\boldsymbol{\theta}_{\mathcal{A}}(t+\eta)}{r_{\mathcal{A}}(t+\eta)}$ is ill-defined. In summary, there are gaps between GF and GD, and our discussion above indicates that the counter term is inevitable to describe the actual dynamics of GD.

## 5.2  Learning Dynamics of Translation-invariant Layers

Next, we apply EoM to translation-invariant layers. To the best of our knowledge, no study analyzes the temporal evolution of translation-invariant layers except for [31] and [32], where only the sum of weights is their focus, while we derive the dynamics of the whole weights. A translation-invariant layer $\mathcal{A}$ is defined as a layer that is invariant under the translation transformation $\psi(\boldsymbol{\theta}, \alpha) := \boldsymbol{\theta} + \alpha \mathbb{1}_{\mathcal{A}}$ ($\alpha \in \mathbb{R}$). For example, a linear layer immediately before the softmax layer is translation-invariant. In the following, we derive EoM and show that its theoretical prediction of decay rates dramatically matches empirical results, indicating the importance of the counter term. In Appendix D, we additionally discuss the differences between GF and GD in translation-invariant layers.

For convenience, we first decompose $\boldsymbol{\theta}_{\mathcal{A}}$ to two vectors (Figure 5); $\boldsymbol{\theta}_{\mathcal{A}\perp}$ is orthogonal to $\nabla f(\boldsymbol{\theta})$, and $\boldsymbol{\theta}_{\mathcal{A}\|}$ is orthogonal to $\boldsymbol{\theta}_{\mathcal{A}\perp}$. Here, note that $\nabla f(\boldsymbol{\theta})$ is orthogonal to $\mathbb{1}_{\mathcal{A}}$ because of translation invariance; in fact, differentiating both sides of $f(\boldsymbol{\theta} + \alpha \mathbb{1}_{\mathcal{A}}) = f(\boldsymbol{\theta})$ with respect to $\alpha$ and setting $\alpha = 0$, we have $\mathbb{1}_{\mathcal{A}} \cdot \nabla f(\boldsymbol{\theta}) = 0$ (see also Lemma A.7 in Appendix A.11). Formally, we define $\boldsymbol{\theta}_{\mathcal{A}\perp}$, $\boldsymbol{\theta}_{\mathcal{A}\|}$, and the projection matrix $P$ as $\boldsymbol{\theta}_{\mathcal{A}\perp} := P\boldsymbol{\theta}_{\mathcal{A}} = \frac{\mathbb{1}_{\mathcal{A}} \cdot \boldsymbol{\theta}_{\mathcal{A}}}{d_{\mathcal{A}}}\mathbb{1}_{\mathcal{A}}$, $\boldsymbol{\theta}_{\mathcal{A}\|} := (I - P)\boldsymbol{\theta}_{\mathcal{A}} = \boldsymbol{\theta}_{\mathcal{A}} - \boldsymbol{\theta}_{\mathcal{A}\perp}$, and $P := \frac{1}{d_{\mathcal{A}}}\mathbb{1}_{\mathcal{A}}\mathbb{1}_{\mathcal{A}}^{\top}$, where $d_{\mathcal{A}}$ is the dimension of $\mathcal{A}$.

We construct the EoM for $\boldsymbol{\theta}_{\mathcal{A}\perp}$ (the EoM for $\boldsymbol{\theta}_{\mathcal{A}\|}$ is given in Appendix D for completeness).

**Theorem 5.2** (EoM for $\boldsymbol{\theta}_{\mathcal{A}\perp}$). *EoM (1) gives $\dot{\boldsymbol{\theta}}_{\mathcal{A}\perp}(t) = -\lambda\boldsymbol{\theta}_{\mathcal{A}\perp}(t) - \eta P\boldsymbol{\xi}(\boldsymbol{\theta}(t))$. Specifically, this is equivalent to $\dot{\boldsymbol{\theta}}_{\mathcal{A}\perp}(t) = -\lambda\boldsymbol{\theta}_{\mathcal{A}\perp}(t) \iff \boldsymbol{\theta}_{\mathcal{A}\perp}(t) = \boldsymbol{\theta}_{\mathcal{A}\perp}(0)e^{-\lambda t}$ for $\boldsymbol{\xi} = \mathbf{0}$ (GF) and $\dot{\boldsymbol{\theta}}_{\mathcal{A}\perp}(t) = -(\lambda + \frac{\eta\lambda^2}{2})\boldsymbol{\theta}_{\mathcal{A}\perp}(t) \iff \boldsymbol{\theta}_{\mathcal{A}\perp}(t) = \boldsymbol{\theta}_{\mathcal{A}\perp}(0)e^{-(\lambda + \frac{\eta\lambda^2}{2})t}$ for $\boldsymbol{\xi} = \tilde{\boldsymbol{\xi}}_0$ (EoM).*

The proof is based on Equations (1, 9) and is given in Appendix A.11. $\boldsymbol{\theta}_{\mathcal{A}\perp}$ monotonically collapses to zero as $t \to \infty$ in either case of $\boldsymbol{\xi} = \mathbf{0}$ or $\boldsymbol{\xi} \neq \mathbf{0}$; thus, as $t$ increases, the dynamics is restricted onto the subspace orthogonal to $\boldsymbol{\theta}_{\mathcal{A}\perp}$ (Figure 5). The decay rate is corrected by the counter term from $\eta\lambda$ to $\eta\lambda + \frac{\eta^2\lambda^2}{2}$, as is also done for $r_{\mathcal{A}}$ in Section 5.1. Therefore, the $\boldsymbol{\theta}_{\mathcal{A}\perp}$ of GD decays faster than that of GF. Figure 6 and Table 1 support our findings. In particular, Table 1 shows that the decay rates predicted by EoM dramatically match those of GD, indicating the importance of the counter term.

Table 1: **Decay rates of $||\boldsymbol{\theta}_{\mathcal{A}\perp}||$.** The theoretical predictions by EoM (third column) dramatically match experimental results of GD (fourth column) much better than GF (second column), indicating the importance of the counter term. LR and WD mean learning rate and weight decay, respectively. The colors correspond to those in Figure 6. See Section 6 for experimental settings.

| (LR, WD) | Theory (GF) | Theory (EoM: **Ours**) | Experiment (GD) |
|---|---|---|---|
| $(10^{-1}, 10^{-2})$ (blue) | $10^{-3}$ | $1.0005 \times 10^{-3}$ | $1.0005003484995967 \times 10^{-3}$ |
| $(10^{-1}, 10^{-3})$ (orange) | $10^{-4}$ | $1.00005 \times 10^{-4}$ | $1.0000500182363355 \times 10^{-4}$ |
| $(10^{-2}, 10^{-2})$ (green) | $10^{-4}$ | $1.00005 \times 10^{-4}$ | $1.0000499809795858 \times 10^{-4}$ |
| $(10^{-2}, 10^{-3})$ (red) | $10^{-5}$ | $1.000005 \times 10^{-5}$ | $1.0000049776814671 \times 10^{-5}$ |
| $(10^{-3}, 10^{-2})$ (purple) | $10^{-5}$ | $1.000005 \times 10^{-5}$ | $1.0000050475312426 \times 10^{-5}$ |
| $(10^{-3}, 10^{-3})$ (yellow) | $10^{-6}$ | $1.0000005 \times 10^{-6}$ | $1.0000005475009833 \times 10^{-6}$ |

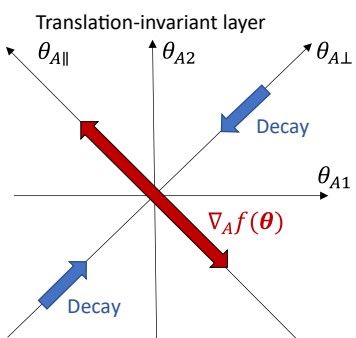

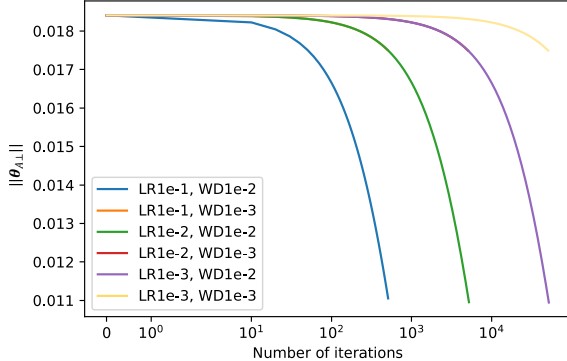

Figure 5: **Learning dynamics of translation-invariant layer.** Here, $\boldsymbol{\theta}_{\mathcal{A}} = (\theta_{\mathcal{A}1}, \theta_{\mathcal{A}2})^\top$. $\boldsymbol{\theta}_{\mathcal{A}\perp}$ decays to $\mathbf{0}$ (also shown in Figure 6). The decay of GD is faster than that of GF (Theorem 5.2). As $t$ increases, the dynamics is restricted onto the subspace orthogonal to $\boldsymbol{\theta}_{\mathcal{A}\perp}$.

Figure 6: **Decay of $\|\boldsymbol{\theta}_{\mathcal{A}\perp}\|$ (GD).** $\|\boldsymbol{\theta}_{\mathcal{A}\perp}\|$ monotonically decays to zero, as suggested by Theorem 5.2. $\mathcal{A}$ is translation-invariant layer. LR and WD mean learning rate and weight decay, respectively. Note that the orange and green curves (LR1e-1, WD1e-3 and LR1e-2, WD1e-2) and the red and purple curves (LR1e-2, WD1e-3 and LR1e-3, WD1e-2) totally overlap. The decay rates of all curves are given in Table 1. See Section 6 for experimental settings.

## 6 Experiment

We explain our experimental settings for Figures 2–6 and Table 1. Our network consists of a first linear layer, swish activation [48], second linear layer, batch normalization [49], third linear layer, and last softmax layer. Cross-entropy is used for the loss function. We note that the second linear layer is scale-invariant, and the last linear layer is translation invariant. The batch normalization uses fixed statistics to keep the scale invariance of the second linear layer. Swish is chosen to ensure differentiability. None of the linear layers have a bias term. The dataset is the training set of MNIST [50], and thus, the batch size is 60,000. Gradient descent is used for the optimizer. We use 64-bits of precision for all computations. To simulate GF and EoM, we use a sufficiently small learning rate ($10^{-5}$). The results are produced from only one random seed to save on computational costs, but we confirm that different random seeds lead to similar results. More detailed information is given in Appendix E and our code. In all experiments, we use $\boldsymbol{\xi} = \tilde{\boldsymbol{\xi}}_0$ for EoM. We do not include higher-order counter terms, such as $\tilde{\boldsymbol{\xi}}_1$, because they require third and higher order derivatives of the loss function and are thus extremely memory-consuming. We could circumvent this issue, e.g., by applying Hessian-free optimization [51], but this is out of our current scope.

## 7 Conclusion and Limitations

In this work, to fill the critical gap between GF and GD, we add a counter term to GF and obtain EoM, a continuous differential equation that precisely describes the discrete learning dynamics of GD. To show to what extent GF and EoM are precise in describing GD's discrete dynamics, we derive the leading order of discretization error, as is often missed in the literature on the continuous approximation of discrete GD algorithms. We further derive a sufficient condition for learning rates for the discretization error to be small. We apply our theory to two specific cases, scale- and translation-invariant layers, indicating the importance of the counter term for a better description of the discrete learning dynamics of GD. Our experimental results support our theoretical findings.

Throughout this paper, we focus only on GD and GF to expose the ideas simply, and our study does not include stochasticity (e.g., SGD and SDE), acceleration methods (e.g., momentum and Nesterov [52]), or adaptive optimizers (e.g., Adam [53]). Nonetheless, they could be combined with our analysis, for example, using error analysis of SDEs [23, 24], continuous-time accelerated methods [7, 54, 9, 13, 14, 55, 16], and continuous-time Adam [56]. See Appendix G for more discussions. Therefore, our study could be extended to import continuous analysis to the discrete analysis of various GD algorithms. In this sense, our work bridges discrete and continuous analyses of GD algorithms.

## Acknowledgment

We thank Shuhei M. Yoshida for his insightful comments on the dynamics of scale-invariant layers and the experimental settings. We also thank Hidenori Tanaka for his discussion that inspired us to start this study.

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
