# OpenReview forum: "Toward Equation of Motion for Deep Neural Networks: Continuous-time Gradient Descent and Discretization Error Analysis"
_NeurIPS.cc/2022/Conference — NeurIPS 2022 Accept_

### Official Review · Reviewer_UB6r · 2022-07-04

**Rating:** 8
**Confidence:** 3
**Soundness:** 4 excellent
**Presentation:** 3 good
**Contribution:** 4 excellent

**Summary:**

This paper is concerned with a theoretical understanding of modelling the dynamics of gradient descent with a differential equation. Previous work (Gradient Flow) describes the differential equation as:

$\frac{d\theta}{dt} = -\nabla_{\theta} L(\theta)$

Which is the Euler discretisation of Gradient Descent:

$\theta_{t+1} = \theta_{t} - \eta \nabla_{\theta}L(\theta_t)$

However, discretisation error exists, such that Gradient Flow and Gradient Descent diverge. This paper derives a counter term to Gradient Flow, labelled by $\xi$:

$\frac{d\theta}{dt} = -\nabla_{\theta} L(\theta) - \eta\xi(\theta)$

The counter term is a functional integral, the paper approximates the counter term with a series solution in $\eta$, with a recursive relationship existing to get from term $k$ to term $k+1$, the new dynamics are called Equation of Motion (EoM). A limit for the learning rate is also derived, which allows accurate simulation of gradient descent using (EoM) with larger step sizes.

Finally these findings are tested on scale-invariant layers and translation-invariant layers and the results support the theoretical findings.

**Questions:**

The few questions/clarifications I have are:

- Line 240 (Definitions), a symmetry is defined as a transformation of parameters such that the loss function is unchanged. Is this definitely correct? Should it not be a transformation that leaves the predictions unchanged?
- Is it possible to run similar experiments for layers that do not have any symmetry, such as a standard linear layer? This could make a nice piece of future work.
- It seems that the majority of experiments were run using the first term in the expansion of $\xi$. Is it possible to run an ablation where we see how adding the next term affects results, and on and on until say the 5th term (to justify using only the first term)? I appreciate this requires huge amounts of memory for the higher derivatives.
- Apart from the theoretical contributions (which are already fantastic), do you envision any practical applications of this work?

**Limitations:**

The authors have been upfront with the limitations of their work. These are given in the conclusion and provide a nice avenue for future research, they are about different optimizers and how using minibatches are not accounted for in the current work. I cannot think of any further limitations.

**Strengths And Weaknesses:**

Strengths:

This is not an area I have significant expertise in, however, overall this paper is very good in my opinion. Specifically:

- The paper shows many impressive theoretical results
- The experiments are extensive and support the theory
- The motivation for the paper is very clear

Weaknesses:

Again, I believe the paper is very good. I think it presents good theoretical results, with sufficient experiment to support this. The only weaknesses overall are in the writing style and presentation, I think the paper is quite math heavy currently, which makes it less accessible. However, this is down to personal preference. For example lines 239-245 (Definitions) feel like quite a complicated way of saying most of the meanings. One definition is $\alpha_{\mathcal{A}} = \alpha I_\mathcal{A} + I_{\mathcal{A}^{C}}$, but it is easier in my opinion to say $\alpha_{\mathcal{A}}$ is $\alpha$ for the parameters in layer $\mathcal{A}$ and $1$ for the others. This can be extended for most of the definitions in this paragraph. I think the best way the paper can be improved is by having as much intuition as possible in the main text, with theorems included, and then possibly having more mathematical detail in the appendix.

Other small points are the presentation of results. Figure captions don't have **Figure n** in line with the caption, but over to the left which feels weird. Table 1 could also be improved, rather than listing the decay rates, it might be more informative to list the differences (and relative differences) in decay rates between GF & GD and EoM & GD.

---

> ### Author Response · Authors · 2022-07-30
> **Reply to Reviewer UB6r**
>
> Thank you for your time and efforts.
> We truly appreciate your perspective on the significance and our contributions.
> We will include all the suggestions in an updated version.
>
>
> > Line 240 (Definitions), a symmetry is defined as a transformation of parameters such that the loss function is unchanged. Is this definitely correct? Should it not be a transformation that leaves the predictions unchanged?
>
> There are mainly two definitions of symmetry of weight parameters: one is the invariance of the output from the network (e.g., [[Godfrey+, 2022](https://arxiv.org/abs/2205.14258)]), and the other is the invariance of the loss function (e.g., [31, 32]). Our definition follows the latter.
>
>
> > Is it possible to run similar experiments for layers that do not have any symmetry, such as a standard linear layer? This could make a nice piece of future work.
>
> Thank you for the suggestion!
> It is interesting to analyze the differences between the dynamics of weight parameters with and without symmetry.
> We expect that
> 1) compared with scale-invariant layers, the decay dynamics of weight norms are dominated by not only weight decay but the loss gradients (in the sense that the weight update contains a lower-order contribution of $\eta$ than scale-invariant layers) if the symmetry is absent, which makes the analysis difficult except for the limiting phase ($t \rightarrow \infty$), and
> 2) compared with translation-invariant layers, not only theta_{A parallel} but theta_{A perp} is affected by the loss gradients if the symmetry is absent.
>
> Thus, we will see clear empirical differences between the layers with and without symmetry.
>
> > It seems that the majority of experiments were run using the first term in the expansion of $\xi$. Is it possible to run an ablation where we see how adding the next term affects results, and on and on until say the 5th term (to justify using only the first term)? I appreciate this requires huge amounts of memory for the higher derivatives.
>
> That is an important experiment to check the validity of using only the first term, but the experiment with the second- and even higher-order terms requires computing higher-order derivatives of the loss function, which is prohibitively heavy (Section 6), as you mentioned.
> Such a heavy computation could be circumvented by applying Hessian-free optimization [51] and any other related techniques, but it is out of our current scope because it requires additional implementations of other technical algorithms.
>
> However, we can infer under what conditions the higher-order corrections cannot be neglected (Figures 2 and 3).
> Figure 2 shows that the higher-order correction dominates the early phase of training.
> Figure 3 shows that we need the higher-order corrections to keep discretization error small for a large learning rate of the order of $\sim 10^{-1}$.
> We avoid these two effects in the experiments in Section 5 by 1) fitting the decay curves in sufficiently long runs to avoid the effects of the early training phase and 2) using small learning rates ($ \ll 10^{-1}$). They make the first-order approximation reliable.
>
> > Apart from the theoretical contributions (which are already fantastic), do you envision any practical applications of this work?
>
> Thank you for the question.
> It would be difficult to directly apply our theory to practical problems, but our work lays foundations for formally quantifying the discrepancy between GF and GD, which is often missing in the literature, and thus bridges discrete and continuous analyses of GD algorithms.
> Such an analysis eventually leads to a deep understanding of the learning dynamics of DNNs, which will be the foundation of new learning algorithms and models for real-world problems.
>
> Thank you again for the support and helpful feedback!

---

> > ### Comment · Reviewer_UB6r · 2022-08-04
> > **Thank you for your responses. Would still like to see another experiment**
> >
> > Dear authors,
> >
> > Thank you for your responses, which answer the majority of my questions well. I would still like to see an experiment on possible the second order term, just to show the difference it can make (which is hopefully small). Maybe this could be done on a very small network? With only a few hundred parameters, just to show the results I think this would improve this paper even more.
> >
> > In light of other reviews placing the work into more context with surrounding literature (thank you to the other reviewers), I have to drop my score slightly. I still recommend accept and think this is a nice paper.

---

> > > ### Author Response · Authors · 2022-08-04
> > > **We have a hard time because...**
> > >
> > > Thank you for your response!
> > >
> > > > I would still like to see an experiment on possible the second order term, just to show the difference it can make (which is hopefully small). Maybe this could be done on a very small network? With only a few hundred parameters, just to show the results I think this would improve this paper even more.
> > >
> > > We tried to run experiments using the second-order term, but we have a hard time removing the OOM error even though we use a network with only a few hundred parameters.
> > > To use the second-order term, we have to compute $\boldsymbol{g}^\top \nabla H \boldsymbol{g}$, $H^2 \boldsymbol{g}$, and $H\boldsymbol{g}$ at a time. The first problem is the computation of $\nabla H$, i.e., the third derivative of the loss function. The second problem is the multiplications of $\nabla H$, $H$, and $\boldsymbol{g}$ (In the experiments in the paper, we addressed this problem by using a numerical differentiation described in Appendix E).
> > > What is worse, the memory consumption multiplicatively increases due to the third problem, namely, the full-batch training.
> > > We will keep addressing these problems.
> > >
> > > The second-order term is $\eta$ $(\sim 10^{-2}, 10^{-3})$ times smaller than the first-order term. Thus, we believe that although adding this term will reduce the discretization error in accordance with our theory, the reduction would be very small.
> > >
> > > Again, thank you for strongly supporting our paper!

---

> > > > ### Comment · Reviewer_UB6r · 2022-08-08
> > > > **That is understandable, maybe even 10 parameters?**
> > > >
> > > > This reasoning is understandable and I accept it. Could it be possible on a network of maybe even 10 parameters, using some sort of synthetic system rather than real data. Maybe it could be a single layer? Or maybe a full network could be trained first, with the first layer being a convolutional layer with possibly 9 parameters. Then freeze all layers except the first, reinitialize the first layer, then train from there?
> > > >
> > > > I understand this is very difficult with the remaining time, let alone the computational/memory costs. But I do think it would make the paper so much stronger, I endorse the paper regardless.

---

> > > > > ### Author Response · Authors · 2022-08-08
> > > > > **It would be nice**
> > > > >
> > > > > We really appreciate your suggestions!
> > > > > Using a tiny synthetic dataset and an extremely small network would be a nice idea.
> > > > > We will keep trying for possible future updates.
> > > > > We agree it would make our paper much stronger.

---

### Official Review · Reviewer_BEXW · 2022-07-11

**Rating:** 7
**Confidence:** 3
**Soundness:** 3 good
**Presentation:** 2 fair
**Contribution:** 3 good

**Summary:**

This paper deals with the discrepancy between the actual discretized gradient descent and its continuous version, i.e., a gradient flow, for describing the equation of motion of learning dynamics more precisely. The discrepancy error is formally introduced by using the backward error in numerical analysis. The authors derive a counter term, which can compensate for such a discrepancy of the gradient flow, thus can describe the actual discretized trajectories in a continuous manner. While the derived counter term is a complicated functional integral equation, it can be analytically solved (for all orders) by assuming the underlying solution is a power series. As an application, the authors use the derived dynamics with the proposed counter term for investigating scaling- and translation-invariant layers.

**Questions:**

As I mentioned previously, it will be nice if the authors can experimentally elaborate on the usefulness of the proposed generalized counter term compared to the first-order counter term.

Is there any idea that can tighten the learning rate bound (13) more efficiently?


**Limitations:**

The authors discuss the limitations of the proposed method, e.g.,  the lack of concerning the mini-batch stochastic GD and other optimizers beyond GD, in Conclusion and Limitations section. It will be nice if the authors also address the questions raised above.

**Strengths And Weaknesses:**

[Note]

Because I am not an expert on learning theory, my evaluation might not be exhaustive. Also, I did not read the proofs in the supplementary material carefully.

[Strengths]

To me, the derived counter term is novel and seems to be useful to predict and interpret complicated learning dynamics of deep neural network models. Although there have been previous studies that incorporate some correction terms with respect to the backward analysis error, to my knowledge they are restricted to 1-st order compensation $\frac{1}{4} \nabla || \nabla f(\theta)||$, which is generally called an implicit gradient regularization. The proposed counter term is generalized for higher orders and can recover the previous studies well as in (9). While the main result (8) seems to be a known technique in the numerical analysis field, I would like to give appropriate credit to the authors for the contribution that introducing such a technique to the deep learning field well.

[Weakenesses]

The authors address a full-batch gradient descent only. The authors also mention such a limitation of this work in Conclusion and Limitations. I am not sure whether the approach can be easily generalized for the mini-batch stochastic gradient descent method.

While the authors theoretically prove high-order corrections are required to cancel the leading order of discretization error, it will be great if the authors (1) experimentally show the discrepancy between the GF with the proposed correction and that with a first order correction, and (2) demonstrate the former can approximate GD well compared the latter, e.g., in Figure 2 or Figure 4.

The proposed method is not guaranteed for GD with a large learning rate, thus cannot be used for explaining some interesting phenomena, e.g., the regularization effect of an initial large learning rate. However, I think it is not a crucial drawback of this paper, considering the essential assumption of GF.

The paper is very dense and thus hard to read. A journal format might be more suitable for a clear representation of this work.

---

> ### Author Response · Authors · 2022-07-30
> **Reply to Reviewer BEXW (1/2)**
>
> Thank you for your time and efforts.
> We agree with the summary and strength given in the review, and we really appreciate your perspective on the significance and novelty of our contributions.
> We will incorporate all the suggestions in an updated version.
>
> > I am not sure whether the approach can be easily generalized for the mini-batch stochastic gradient descent method.
>
> There are a few gaps to generalize our approach to the mini-batch SGD (and other practical optimizers).
> An interesting approach is to extend the error analysis of the Euler-Maruyama method to include the counter term.
> The first-order error analysis is given in [23, 24], and a general error analysis between SGD and SDE is given in [67]. They can be a good starting point.
>
> In addition, several recent papers [7, 9, 13, 14, 16, 54, 55, 56] proposed variants of gradient flow, and we believe our work lays foundations for formally quantifying the discrepancy between these variants and the practical optimizers they aim to represent.
>
> We mention these points in Section 7 and Appendix G.
>
> > While the authors theoretically prove high-order corrections are required to cancel the leading order of discretization error, it will be great if the authors (1) experimentally show the discrepancy between the GF with the proposed correction and that with a first order correction, and (2) demonstrate the former can approximate GD well compared the latter, e.g., in Figure 2 or Figure 4.
>
> Thank you for the constructive suggestion!
> Yes, it is definitely interesting to experimentally compare the first-order correction with the higher-order corrections.
> Unfortunately, we use only the first-order correction in the experiment because the higher-order corrections require the third- and higher-order derivatives of the loss function and are extremely memory-consuming (Section 6).
> Such a heavy computation could be circumvented by applying Hessian-free optimization [51] and any other related techniques, as discussed in Section 6, but it is out of our scope because it requires additional implementations of other technical algorithms.
> Instead, we focus on theoretical analysis of our approach and on experiments in a minimum setting for proof of concept.
> Our experimental settings highlight the differences between the counter term equal to zero and the non-zero counter term, using only the first-order correction (Section 4.1).
>
> Please note that we can infer the effect of the higher-order corrections in the experiment only with the first-order correction.
> We discuss in Section 4.1 (Figure 2) that the higher-order corrections dominate the early phase of training, and therefore we can expect that the presence of the higher-order corrections approximates GD better even in the early phase of training.

---

> > ### Author Response · Authors · 2022-07-30
> > **Reply to Reviewer BEXW (2/2)**
> >
> > > The proposed method is not guaranteed for GD with a large learning rate, thus cannot be used for explaining some interesting phenomena, e.g., the regularization effect of an initial large learning rate. However, I think it is not a crucial drawback of this paper, considering the essential assumption of GF.
> >
> > We agree that the analysis of GD's learning dynamics with a large learning rate is an exciting future work.
> > We are aware that our analysis cannot be applied to the large-learning-rate regime because we assume learning rates must be sufficiently small.
> > This condition is hard to avoid because, as you mentioned, it is the essential assumption of general GF-type methods.
> >
> > One approach is to include as many higher-order corrections to EoM as possible, which prevent discretization error from diverging.
> > However, this approach is not relevant when the step size is as large as $O(1)$ where the series expansion with respect to the step size diverges.
> >
> > There are interesting phenomena in the large-learning-rate regime: unstable convergence with large learning rates, escaping from local minima, and the regularization effect of an initial large learning rate.
> > We do not have any other concrete and relevant ideas to tackle these interesting problems so far, but discussions are welcome!
> >
> >
> > > Is there any idea that can tighten the learning rate bound (13) more efficiently?
> >
> > That is a very interesting question.
> > Basically, it may be difficult to tighten the bound in our setting because this bound is deeply related to the error bound of the classic Euler method. Tightening the bound may be possible by restricting the conditions on the objective function, e.g., assuming convexity.
> >
> > ---
> > In light of our responses, we would greatly appreciate it if you would be able to consider a raise in your score. Thank you!

---

> > > ### Comment · Reviewer_BEXW · 2022-08-08
> > > **Reply to the authors**
> > >
> > > I appreciate the authors’ thorough response to my questions. I still think the paper is worthy of publication, thus keep my score (7) with a higher confidence.

---

> > > > ### Author Response · Authors · 2022-08-08
> > > > **Reviewer BEXW**
> > > >
> > > > Thank you for your support!

---

### Official Review · Reviewer_2BkU · 2022-07-11

**Rating:** 6
**Confidence:** 4
**Soundness:** 3 good
**Presentation:** 3 good
**Contribution:** 2 fair

**Summary:**

The authors derive an Equation of Motion, i.e., a continuous differential equation that matches the discrete time dynamics of gradient descent more clearly. They do so by adding a counter terms to Gradient Flow that cancels out higher order discretization errors in DNNs, and this counter term is derived using backward error analysis, more precisely it is the solution to Equation 6 in the paper.

Given they are using backward error analysis, they are also able to quantify the discretization error for GD approximation of GF along with the counter term, and hence also provide a bound on the learning rate such that this discretization error is small.

The authors apply their Equation of Motion to translation and scale invariant layers and show that their theoretical predictions better match GD.

**Questions:**

In the current state the results seem to be very similar to the previous work (as I have mentioned in the previous comment), and besides the series expansion in Theorem 3.3, it is very hard to adjudge the precise novelty of the work, esp since most of the results discussed in the next sections assumes that the counter term is either 0 or they assume the first order counter term. I would really appreciate if the authors could clarify their contribution.

Another minor comment, I think it would be easier to read the paper (esp the proofs) if the author provided a sketch after the main statement in the main paper as well as in the appendix.

**Limitations:**

Yes the authors have discussed the limitations of their work.

**Strengths And Weaknesses:**

My main concern is that this paper does not provide any interesting new result. The main novelty of the paper is the general form of the counter term (which is derived in Theorem 3.3), as opposed to previous work for example Barrett and Dherin (Implicit gradient regularization) which only uses the first order term as their regularization term.

While the authors mention that they derive the discretization error (in Corollary 4.1), the precise formulation is not provided and the rate is provided as an upper bound (using Big-OH) which I believe is an artifact of standard series expansion results.

Furthermore, for most of the results on the discretization error bounds and the upper bound on the learning rate, the authors assume that the counter term is either equal to zero or assume the first order counter term (i.e., the term in equation 9).

With the counter term equal to 0, the analysis matches a lot of the previous work (example Elkabetz and Cohen, [18]) and with the first order term the main theoretical results are very similar to the ones already established in Barrett and Dherin (who introduce the first order counter term as the implicit regularizer and the error is discussed in Theorem 3.1)

Besides this, the authors do apply their analysis to characterize the learning dynamics of scale and translation invariant layers and show that with the inclusion of the first order (adding higher order counter term is going to be computationally expensive) they are able to better predict the decay of parameter norm, which is interesting but not that surprising.

---

> ### Author Response · Authors · 2022-07-30
> **Reply to Reviewer 2BkU (1/2)**
>
> We thank the reviewer for their time and efforts.
> We will incorporate all the suggestions in an updated version.
>
> > My main concern is that this paper does not provide any interesting new result.
>
> We respectfully disagree with this point.
> Let us clarify our contributions and novelty and solve your concerns below.
>
>
> > The main novelty of the paper is the general form of the counter term (which is derived in Theorem 3.3), as opposed to previous work for example Barrett and Dherin (Implicit gradient regularization) which only uses the first order term as their regularization term.
>
> Thank you for appreciating the novelty. Yes, it is one of our contributions.
>
> > While the authors mention that they derive the discretization error (in Corollary 4.1), the precise formulation is not provided ...
>
> Please see Appendix A.4, where we provide the precise formulation (proof) of Corollary 4.1.
>
> > ... (in Corollary 4.1) ... the rate is provided as an upper bound (using Big-OH) which I believe is an artifact of standard series expansion results.
>
> Could you please clarify what you mean by ``an artifact of standard series expansion results''?
> We are open to discussion!
>
>
> > Furthermore, for most of the results on the discretization error bounds and the upper bound on the learning rate, the authors assume that the counter term is either equal to zero or assume the first order counter term (i.e., the term in equation 9).
>
> We use the counter term equal to zero and the first-order counter term here (Section 4.2 and Appendix A.6) in order to 1) compare bounds with a previous study [18] (Corollary 4.2) and 2) confirm that the counter term improves the bound compared with the counter term equal to zero (Corollary A.1).
> Note that these results can be easily extended to higher-order terms, as is mentioned at the end of Section 4.2. Specifically, the bound is $\eta < O((\epsilon / k)^{1/(\gamma+2)})$ when we use a $O(\eta^\gamma)$ counter term.
>
>
> > With the counter term equal to 0, the analysis matches a lot of the previous work (example Elkabetz and Cohen, [18]) and with the first order term the main theoretical results are very similar to the ones already established in Barrett and Dherin (who introduce the first order counter term as the implicit regularizer and the error is discussed in Theorem 3.1)
>
> With the counter term equal to zero, our analysis DOES NOT exactly match [18].
> The comparison regarding this point is provided in Lines 223--228.
> Specifically, the bound given in [18] includes factors that depend on both the step size and the smoothness of the loss, which make it difficult to separate the dependence of the bound on the step size, while our bound has an explicit and separate dependence on the step size and the smoothness of the loss (Corollary 4.2).
>
> With the first order term, our theoretical results (Corollary 4.1) have crucial differences from Theorem 3.1 in Barrett & Dherin [27]: 1) [27] is a local error analysis (one-step discretization error), while ours is a global error analysis (whole-steps discretization error), 2) [27] gives only the order of local error with respect to the step size, while we provide the pre-factors of the step size as well (Corollary A.1), 3) [27] is the first-order error analysis, while ours can be extended to the all-order error analysis.
>
> Therefore, our results provide independent contributions compared with [18] and [27].

---

> > ### Author Response · Authors · 2022-07-30
> > **Reply to Reviewer 2BkU (2/2)**
> >
> > > the authors do apply their analysis to characterize the learning dynamics of scale and translation invariant layers and show that with the inclusion of the first order (adding higher order counter term is going to be computationally expensive) they are able to better predict the decay of parameter norm, which is interesting but not that surprising.
> >
> > Let us clarify our contributions in Section 5, where we applied EoM to scale-invariant layers and translation-invariant layers.
> > - We showed that a modification to the decay rate of the weight norm given in [33] is needed because [33] ignores discretization error.
> > - We also showed that GF cannot reproduce the limiting dynamics ($t\rightarrow\infty$) of the weight norm and angular update of scale-invariant layers unless the counter term is included; in other words, we explicitly provide a counterexample that GF cannot fully explain the learning dynamics of GD due to discretization error. Therefore, our work sheds light on the importance of discretization error, which is often missing in the literature on continuous-time GD. More discussions on the differences between GF and GD are given in Appendix C and D.
> > - We are the first to derive the dynamics of the whole weights in translation-invariant layers, which are equipped with almost all the networks that use the softmax loss.
> > we empirically showed that our theoretical prediction of decay rates dramatically matches empirical results (Table 1). Notably, Table 1 shows that our theoretical prediction captures the differences of the order of $5 \times 10^{-7} \sim 5 \times 10^{-10}$.
> > There are few works that conduct such a precise experiment in deep learning, to our knowledge.
> >
> > ---
> > In view of our responses, we would greatly appreciate it if you would be willing to consider raising your score. Thank you very much!

---

> > ### Comment · Reviewer_2BkU · 2022-08-06
> > **Thank you for the clarifications**
> >
> > I thank the authors for the clarifications and the details response delineating the differences with the previous work. With this in mind, I will update my score.

---

> > > ### Author Response · Authors · 2022-08-07
> > > **To Reviewer 2BkU**
> > >
> > > Thank you very much!

---

### Official Review · Reviewer_NdAp · 2022-07-18

**Rating:** 6
**Confidence:** 2
**Soundness:** 3 good
**Presentation:** 4 excellent
**Contribution:** 3 good

**Summary:**

This paper derives a counter term to the gradient flow ODE formulation that reduces the discretization error from Euler's method, which is gradient descent. When this correction term is expanded as a Taylor series, adding a select number of terms reduces the discretization order accordingly. This is then used to analyze the behavior of GD under symmetry constraints, specifically scale- and translation-invariant parameters. Specifically, this adds learning-rate-dependent correction terms to the decay rates of certain quantities, which matches gradient descent in practice.

**Questions:**

Can you clarify the connection between this correction term and the global truncation error of Euler's method (i.e. a bound on e_k)? I expected them to look very similar but seems to not be the case.

Depending on the step size, Euler's method can result in inconsistencies where the discretization completely diverges from the gradient flow, in particular when the step size is much larger than the Lipschitz constant of g, and results in a discretization that cannot be meaningfully modeled by any ODE. I imagine this would correspond to the correction term becoming infinite and the zero-th order approximation to the correction term becoming somewhat meaningless. Does the equation of motion reflect this kind of behavior in any way?

Can the correction term be used to correct the drift of the SDE formulation to stochastic gradient descent?

Can Corollary 4.2 be experimentally verified?

**Limitations:**

.

**Strengths And Weaknesses:**

Pros:
  - Quite an interesting take which attempts to correct the theoretical ODE formulation in order to match practice (as opposed to bridging this gap by using higher order solvers, for example).
  - The motivation, process, and theoretical results are presented very well. I could follow understand every result (just the results; not the proofs) despite not being an expert in the theory of gradient descent.

Cons:
  - I imagine the theory (of estimating discretization error of ODEs) has been done before, though perhaps not in this exact context. The zero-th order term (Eq 9) has shown up in machine learning studies before.
  - It's not clear to me if we gain anything from using the higher order terms in the derivation, as the analysis and experimental results all use only the zero-th order term.

---

> ### Author Response · Authors · 2022-07-30
> **Reply to Reviewer NdAp (1/2)**
>
> We thank the reviewer for their time and efforts, as well as their valuable comments.
> Below, we address the comments and questions raised in the review.
>
> > I imagine the theory (of estimating discretization error of ODEs) has been done before, though perhaps not in this exact context.
>
> We discuss this point in Section 3.3.
>
> Equation (8), which gives the discretization error in accordance with Corollary 4.1, can be found in [35] as a higher-order backward error analysis (different context from ours, though). However, our derivation has independent contributions: 1) we clarify that the counter term cancels the leading order of discretization error (Theorem 3.2), and 2) we find that the discretization error itself is also given by the counter term (Corollary 4.1).
>
> > The zero-th order term (Eq 9) has shown up in machine learning studies before.
>
> We also discuss this point in Section 3.3.
>
> Equation (9) often appears in the literature on backward error analysis [21, 35] and its related topics in machine learning, e.g., [23, 24, 27, 28, 31, 41].
> Typically, Equation (9) is added to gradients in continuous equations (e.g., SDE) to close the gap between continuous equations and discrete algorithms (e.g., SGD) by canceling (at least **first-order**) discretization error.
> However, **higher-order** discretization error is neglected in these studies.
> In contrast, our solution (Equation (8)) cancels **all** orders of discretization error.
>
> > It's not clear to me if we gain anything from using the higher-order terms in the derivation, as the analysis and experimental results all use only the zero-th order term.
>
> **Not** all the analysis and experimental results use the zeroth-order term.
> Let us clarify why we used the zeroth-order term in a part of our paper and summarize our contributions related to the higher-order terms.
>
> Most of the texts up to Section 4.1 (except for the experiment) are dedicated to the all-order counter term. Equations (9) (zeroth), (10) (first), and (12) (zeroth) are presented to simply give an intuition to complicated equations by showing simple low-order examples.
> At the end of Section 4.1, we demonstrate that the higher-order terms dominate the early phase of training.
>
> In Section 4.2 and Appendix A.6, we provide the learning rate bounds for 1) the counter term equal to zero and 2) the zeroth-order counter term in order to 1) compare bounds with a previous study [18] and 2) confirm that the counter term improves the bound.
> These results are not limited to these low-order cases but can be easily extended to higher-order terms, as is mentioned at the end of Section 4.2.
>
> In Section 5 and experiments, we use the zeroth-order term because 1) we would like to simplify the analysis, 2) we need to avoid computing higher-order derivatives of the loss function (e.g., the derivative of a Hessian) to simulate EoM, which is prohibitively memory-consuming, and 3) we can at least see the differences between GF with and without the counter term (mentioned at the beginning of Section 5) even when we only use the zeroth-order term.
> However, please note that the zeroth-order term dominates the counter term and thus is sufficient to analyze the effect of the counter term when the step size is small.
>
> In Appendix G, we provide a higher-order correction to the decay rate of scale-invariant layers, which implies that the higher-order corrections increase the decay rate and allow it to approach GD's decay rate.
>
> Our paper reveals the above findings from the higher-order terms.

---

> > ### Author Response · Authors · 2022-07-30
> > **Reply to Reviewer NdAp (2/2)**
> >
> > > Can you clarify the connection between this correction term and the global truncation error of Euler's method (i.e. a bound on $e_k$)? I expected them to look very similar but seems to not be the case.
> >
> > A well-known global truncation error of Euler's method is $|| {\boldsymbol{e}}_k || \leq \frac{c \eta}{L}(e^{Lt}-1)$, where $\eta$ is the step size, $L$ is the Lipschitz constant, and $c$ is a constant related to the second derivative.
> > The proof is strongly based on Lipschitz smoothness.
> > If we use the Taylor series instead of it, we can obtain $|| {\boldsymbol{e}}_k || \leq k\eta^2 C + O(\eta^3)$ (Equation (50) in Appendix A.5), where $C$ is a constant related to the smoothness of the trajectory.
> > This is what is done in our proof of Corollary 4.2 (Appendix A.5).
> > Though these two bounds are not exactly the same, they share several characteristics; the bound becomes large when 1) $k$ (or $t$) is large, 2) the objective function is non-smooth, and 3) the step size is large.
> >
> > > Depending on the step size, Euler's method can result in inconsistencies where the discretization completely diverges from the gradient flow, in particular when the step size is much larger than the Lipschitz constant of g, and results in a discretization that cannot be meaningfully modeled by any ODE. Does the equation of motion reflect this kind of behavior in any way?
> >
> > Yes. Please see Figure 3, where we show that the discretization error between GF and EoM blows up for a large learning rate.
> > This result is consistent with our theoretical findings (Corollary A.1).
> > We can expect that this divergence becomes less likely when we add higher-order counter terms to EoM (Corollary 4.1).
> >
> > > Can the correction term be used to correct the drift of the SDE formulation to stochastic gradient descent?
> >
> > Thank you for the question.
> > It is exactly a part of our future work discussed in Section 7 and Appendix G, where we cite [23, 24, 67].
> > An interesting approach is to extend the error analysis of the Euler-Maruyama method to include the counter term.
> > The lowest-order error analysis is given in [23, 24], and a general error analysis between SGD and SDE is given in [67], and thus they can be a good starting point.
> >
> > In addition, several recent papers [7, 9, 13, 14, 16, 54, 55, 56] proposed variants of gradient flow, and we believe our work lays foundations for formally quantifying the discrepancy between these variants and the practical optimizers they aim to represent.
> > We discuss these points in Section 7 and Appendix G as well.
> >
> > > Can Corollary 4.2 be experimentally verified?
> >
> > Yes. Please see Figures 3 and 4.
> > They are consistent with Corollary 4.2; i.e., 1) the discretization error blows up for a large learning rate ($\eta=10^{-1}$ in Figure 3), 2) it increases as the number of steps increases (Figure 4), and 3) most of it is produced in the early phase of training, where the objective function tends to be non-smooth, and the gradients tend to be large.
> >
> > ---
> > In light of our responses, we kindly ask that you consider raising your score. Thank you very much!

---

### Author Response · Authors · 2022-07-30
**General message to Reviewers**

We thank the reviewers for their careful reading to appreciate the strengths of the paper:
- Excellent presentation [NdAp]
- Excellent soundness [UB6r]
- Excellent contribution [UB6r]
- Interesting take which attempts to correct the theoretical ODE formulation in order to match practice [NdAp]
- Novelty of the general form of the counter term [2BkU]
- The derived counter term is novel and seems to be useful to predict and interpret complicated learning dynamics of deep neural network models [BEXW]
- Able to better predict the decay of parameter norm, which is interesting [2BkU]
- Contribution that the authors introduce a technique in the numerical analysis field to the deep learning field [BEXW]
- The experiments are extensive and support the theory [UB6r]
- The paper shows many impressive theoretical results [UB6r]
- The motivation for the paper is very clear [UB6r]

Please find our official comments in each of the review threads.

We are looking forward to discussing our paper with all of you!

---

### Meta-Review · Area_Chair_4E9a · 2022-08-24

**Recommendation:** Accept
**Confidence:** Certain

**Metareview:**

Reviewers were unanimous in recommending that the paper be accepted, and I accordingly recommend the same.  I encourage the authors to take into account suggestions made by reviewers so as to further improve the text in the camera-ready version.

**Award:**

No

---

### Decision · Program_Chairs · 2022-09-14

Accept